# A Reversible Low Frequency Alternating Current Nerve Conduction Block Applied to Mammalian Autonomic Nerves ^†^

**DOI:** 10.3390/s21134521

**Published:** 2021-07-01

**Authors:** M. Ivette Muzquiz, Landan Mintch, M. Ryne Horn, Awadh Alhawwash, Rizwan Bashirullah, Michael Carr, John H. Schild, Ken Yoshida

**Affiliations:** 1Department of Biomedical Engineering, Indiana University-Purdue University Indianapolis, Indianapolis, IN 46202, USA; mimuzqui@alumni.iu.edu (M.I.M.); hornm@iupui.edu (M.R.H.); jschild@iu.edu (J.H.S.); 2Ripple LLC, Salt Lake City, UT 84106, USA; Landan@rppl.com; 3Weldon School of Biomedical Engineering, Purdue University, West Lafayette, IN 47907, USA; aalhawwa@purdue.edu; 4Biomedical Technology Department, King Saud University, Riyadh 11362, Saudi Arabia; 5Galvani Bioelectronics Inc., Collegeville, PA 19426, USA; rizwan.x.bashirullah@galvani.bio (R.B.); michael.j.carr@galvani.bio (M.C.)

**Keywords:** neuromodulation, nerve conduction block, reversible nerve block, low frequency alternating current block

## Abstract

Electrical stimulation can be used to modulate activity within the nervous system in one of two modes: (1) Activation, where activity is added to the neural signalling pathways, or (2) Block, where activity in the nerve is reduced or eliminated. In principle, electrical nerve conduction block has many attractive properties compared to pharmaceutical or surgical interventions. These include reversibility, localization, and tunability for nerve caliber and type. However, methods to effect electrical nerve block are relatively new. Some methods can have associated drawbacks, such as the need for large currents, the production of irreversible chemical byproducts, and onset responses. These can lead to irreversible nerve damage or undesirable neural responses. In the present study we describe a novel low frequency alternating current blocking waveform (LFACb) and measure its efficacy to reversibly block the bradycardic effect elicited by vagal stimulation in anaesthetised rat model. The waveform is a sinusoidal, zero mean(charge balanced), current waveform presented at 1 Hz to bipolar electrodes. Standard pulse stimulation was delivered through Pt-Black coated PtIr bipolar hook electrodes to evoke bradycardia. The conditioning LFAC waveform was presented either through a set of CorTec^®^ bipolar cuff electrodes with Amplicoat^®^ coated Pt contacts, or a second set of Pt Black coated PtIr hook electrodes. The conditioning electrodes were placed caudal to the pulse stimulation hook electrodes. Block of bradycardic effect was assessed by quantifying changes in heart rate during the stimulation stages of LFAC alone, LFAC-and-vagal, and vagal alone. The LFAC achieved 86.2±11.1% and 84.3±4.6% block using hook (N = 7) and cuff (N = 5) electrodes, respectively, at current levels less than 110 µA_p_ (current to peak). The potential across the LFAC delivering electrodes were continuously monitored to verify that the blocking effect was immediately reversed upon discontinuing the LFAC. Thus, LFACb produced a high degree of nerve block at current levels comparable to pulse stimulation amplitudes to activate nerves, resulting in a measurable functional change of a biomarker in the mammalian nervous system.

## 1. Introduction

The peripheral nervous system (PNS) is a medium that the body uses to communicate, control, and relay information within itself. Electrical signals are conveyed to and from the central nervous system, the brain and spinal cord, to muscles, major organs, and sensors embedded throughout the body using action potential (AP) nerve impulses coursing along the neural axonal pathways of the PNS. These signals are used not only to convey sensations and control muscles, but also used to control and regulate the function of major organs such as the kidneys, pancreas, liver, and heart within the autonomic branch of the PNS, the Autonomic Nervous System (ANS). In the case of chronic diseases, the pathology can alter the set point and control system of organs and organ systems, resulting in illness and abnormal function. Electrical stimulation of the ANS has long held hope as a means to modulate the information within the nervous system to help restore function to dysfunctional or poorly functioning organs.

Electrical stimulation can be used to alter the neural information flow and signaling broadly speaking in two modes: Activation or Block. Activation is the mode through which electrical pulses are used to excite and generate nerve activity. It can be thought of as a means to introduce or add to the neural traffic. The activation of nerve fibers using electrical discharges has been known since antiquity and understood for at least a century [2]. Various forms of the standard rectangular pulse stimulus waveform with charge balancing counter waveforms are currently commonly used. The degree of activation is controlled by modulating the strength of stimulation, controlled by the duration or amplitude of the pulse, the Pulse Width (PW) or Pulse Amplitude (PA). When applied to peripheral nerves the added nerve activity comes in the form of a compound action potential volley that is synchronous to the stimulation pulse and propagates away from the stimulating electrode without regard to the normal directionality of the nerve fiber.

Block refers to methods to locally extinguish the neural traffic transiting within the nerve through the site of electrical current application thus preventing transmission of action potentials. Methods to block propagating APs using electrical stimulation are a more recent discovery and a current topic of investigation. Many of the current methods being used or investigated can be effective include the following: Direct current (DC) block [3,4,5], kilohertz frequency alternating current block (KHFACb) [6,7,8,9], anodal block [10,11,12], collision block [13], and quasi-trapezoidal stimulation [14,15,16,17]. Although translation to clinical use is forthcoming some methods have associated drawbacks that need resolution before translation.

The ability of an electrode interface to sustain an electrical current relies on Faradaic and non-Faradaic processes. The composition of these two general types of currents project to the charge carrying capacity of the interface which depends heavily upon the chemoelectric reaction pathways available to the electrode material. It is, thus, highly surface chemistry and material dependent. DC and low frequency currents are primarily sustained by Faradaic reactions, while higher frequency currents are sustained by non-Faradaic processes such as the charging and discharging of the double layer capacitance. Electrochemical impedance spectroscopy and the electrode impedance at low frequencies can give an indication of whether an electrochemical pathway exists. If a pathway exists, the low frequency impedance would have a resistive characteristic, while if it does not exist, the electrode impedance will retain a capacitive impedance characteristic at low frequencies. However, even if an electrode has a resistive low frequency impedance, there is a limit to the ability of the electrode to maintain the current demands. Exceeding the charge carrying capacity of a permissive pathway results in the need for more potential to maintain the current, polarizing the electrode. Thus, a linear input-output relationships between current and electrochemical cell potential can be an indication that the electrode is operating reversibly within its charge carrying capacity. Maintaining a constant or low frequency forward current represents electrochemically driving a Faradiac reaction in the forward direction and producing and accumulating reaction products that, depending upon the chemistry of the interface, could be but not necessarily toxic to biological tissues. Reversing the current to reverse the charge injected onto the interface is an important method to reverse the electrochemical reactions and maintain the interface. In the case of DC block, a ramp or DC current is passed through the blocking electrode. This continuous charge injection represents an accumulation of charge at the electrode interface resulting in an imbalance [18], and the potential production of toxic electrochemical byproducts. Depending upon the rate of accumulation of these byproducts, the nerve can be quickly damaged. A more recent method uses a sinusoidal charged balanced waveform in the frequency range from 1 kHz to 40 kHz, the kilohertz frequency alternating current block (kHFACb). Since the waveform is zero mean with with short cycle durations, the charge reverses itself and there is no net accumulated charge to avoid the possibility of nerve damage [6,7,8,9]. However one drawback that has been reported is an associated ‘onset response’ [7,19]. The onset response is characterized by transient nerve activation that can last for several seconds. This can result in a transient sensation, unintended activation of the target organ, or muscle contractions. Efforts have been made to mitigate the effects from DC block and kHFACb by generating a blocking waveform that is the combination of the two [20,21,22,23] Charge balanced direct current (CBDC) carousel block is a method that mitigates the onset activation where a ramped DC pulse or trapezoidal pulse is used to achieve block which is charge balanced with a long but equal charged discharging phase. In addition, high capacitance materials (Pt Black) is used to prevent reactive species from being produced [24]. Mitigation comes at the cost of complexity. Nevertheless, there can still be a concern of possibility of nerve damage since the method requires the use of various electrodes and current sources at levels in the milliamps range. The low frequency alternating current blocking (LFACb) waveform described in the present paper is a simple pure tone sinusoidal waveform defined by its amplitude and frequency. In our pilot work, while investigating kHFACb in earthworm nerve cords, we discovered that phasic blocking of action potentials could be achieved by reducing the frequency of the sinusoidal waveform to <10 Hz. When a continuous train of stimuli was presented to the nerve, a discontinuous action potential train emerged. Only the action potentials that transited the blocking electrode while the sinusoidal waveform was near its zero crossing emerged from the blocking electrode. There was a blocking phase when the sinusoid was above blocking threshold and a non blocking phase when the sinusoid was below the blocking threshold. The waveform was tested on excised canine and porcine vagus nerves and the blocking phenomenon was found to be conserved across species (canus–sus) and class (clitellata–mammalia) [25]. With further investigation, nerve conduction block was achieved at approximately half of the current levels that was required for kHFACb and was achieved within the electrochemical water window of the Pt-Black coated bipolar electrodes. The electrochemical water window is defined as the water reduction and oxidation voltage limits within which electrolysis is not likely to occur and electrodes would operate linearly [26,27]. Unlike kHFACb, LFACb in our pilot work showed no indication of onset activation. The low thresholds of LFACb are comparable to those of DC blocks. However, the sinusoidal charge balanced waveform prevents the accumulation of unbalanced charge.

The present work aims to build upon the prior work in the earthworm and excised mammalian nerves to ascertain whether LFACb can achieve a physiologically relevant level of nerve block. Using the bradycardic effect elicited by conventional pulse stimulation of the vagus nerve as a biomarker, we sought to assess and demonstrate the LFAC waveform applicability and determine whether using Pt-Black coated hook or PEDOT based Amplicoat^®^ coated cuff electrodes has an impact on the degree of nerve block.

## 2. Materials and Methods

### 2.1. *In-Vivo* Surgical Preparation

This work was conducted under animal use protocols reviewed and approved by the IUPUI School of Science Institutional Animal Care and Use Committee (SoS IACUC) at Indiana University-Purdue University Indianapolis (IUPUI). All methods were in compliance with institutional and governmental guidelines and regulations. The study used a total of 12 adult (162–391 g) Sprague-Dawley rats of mixed gender. The electrophysiological preparation followed Cruz et al. [28] to instrument the animal and access the cervical vagus nerve, and [1]. Briefly, the animals were anaesthetised to the surgical plane following induction using Isoflurane (Vedco Inc., St. Joseph, MO, USA), followed by an intraperitoneal (IP) injection (0.8 mL/100 g) of a compounded solution of urethane (800 mg/kg, Sigma-Aldrich Co., St. Louis, MO, USA) and alpha-chloralose (80 mg/kg, Acros Organics, Fair Lawn, NJ, USA). Anaesthesia at the surgical plane was maintained using supplemental IP injections of the compounded urethane/alpha-chloralose solution. Once anaesthetized, body temperature was maintained using a heating pad (HTP-1500 and ST-017 Soft-Temp Pad, Adroit Medical Systems, Loudon, TN, USA). Blood pressure was monitored using a heparinized saline (30 U/mL) filled PE-100 tube catheter placed in the left femoral artery. The artery was surgically exposed and a short length (10 mm) of the catheter was inserted and secured to the artery. Systemic arterial blood pressure (BP) was monitored through this access via a calibrated pressure transducer (#159905, Radnoti LLC, Covina, CA, USA) connected to the data acquisition system. Access to the left carotid artery and left cervical vagus was obtained through a midline incision on the ventral side of the neck followed by blunt dissection to visualize the structures. A tracheotomy tube was inserted through an incision in the trachea to ready facilitation through mechanical ventilation (Model 683, Harvard Apparatus, Holliston, MA, USA) in the event the animal stopped breathing. The bio-marker of interest, ECG, was monitored continuously using needle electrodes on the chest. The raw ECG signal was bandpass filtered (0.1–300 Hz) and amplified (1000×) using a differential amplifier with active headstage (DP-311, Warner Instruments, Hamden, CT, USA) before the signal was passed to the data acquisition system.

### 2.2. Electrode Placement

Two bipolar extrafascicular electrodes were placed on the exposed left cervical vagus shown in Figure 1. A Pt-Ir Bipolar hook electrode with 800 µm anode/cathode spacing (PBAA0875, FHC, Bowdoin, ME), electroplated with Pt-Black was used for vagal pulse stimulation. The LFACb waveform was delivered via a Pt-Black coated Pt-Ir hook electrode for 7 experiments. For the remaining 5 experiments, a 0.5 mm I.D. bipolar cuff with 1.0 mm contact pitch and 0.5 mm contact width (1041.5008.01,CorTec GmbH, Freiburg, Germany) was used. The CorTec cuff electrode came with Pt contacts coated with a commercial PEDOT (Amplicoat^®^, Heraeus Medical Components) to improve stimulation and sensing performance of the electrode [29,32]. The Amplicoat^®^ or Pt-Black coated electrodes hook electrodes provided electrodes with a sufficiently low, low frequency impedance to keep the cell potential (the potential across the LFAC bipolar electrode pair) within the water window. The rostral hook electrode (RE) was used for vagal stimulation. Meanwhile the caudal electrode (CE), hook or cuff, was used to deliver the conditioning waveform. The left cervical vagus was crushed and ligated rostral to the stimulating hook electrode to eliminate the visceral afferents as the origin of bradycardia. The right vagus nerve was remained intact to maintain autonomic reflexes for stability.

### 2.3. Nerve Stimulation and Experimental Paradigm

A standard rectangular pulse stimulation consisted of a train of 10 pulses (0.1, 1.0 or 2.0 ms pulse width, 25 Hz pulse frequency, 200 ms train duration, and 170 ms train delay) was applied to the vagus nerve using a opto-isolated stimulator (DS3, Digitimer Ltd., Hertfordshire, UK) triggered by a pulse generator (33120A, Hewlett Packard, Engelwood, CO, USA) at an adequate level to evoke bradycardia. The stimuli were delivered at 2 Hz to the RE and in-phase with peaks of the sinusoid due to the observed blocking effect in and in silico [25,30]. The stimuli were titrated to cause only a visible change in ECG and BP without resulting in the animal crashing. The vagal stimulus, applied without block, causes the heart rate to drop from ∼5 Hz to ∼1 Hz (∼300 beats per minute (BPM) to ∼60 BPM). This decrease in HR causes an acute drop in mean blood pressure from 90–110 mmHg to less than 50 mmHg. When the BP dropped below ∼50 mmHg, vagal stimulation was discontinued to enable it to return to its normal set point.

The LFACb waveform was generated using a dual channel waveform generator (DG5072, Rigol Tech, Beaverton, OR, USA) followed by an isolated voltage controlled current source (CS580, Stanford Research Systems, Sunnyvale, CA, USA). Although the CS580 current source has an isolated (floating) output mode, using the voltage monitor defeats the isolation. Thus, a custom built linear isolator based around a Burr-Brown ISO120 was used to maintain electrical isolation. Maintaining electrical isolation provided systemic protection to the animal and minimized voltage transients and cross-talk. It also ensured the electrical isolation between the pulse stimulator and LFAC stimulator to maintain independence of currents presented to their respective electrodes without interaction. Adequate block amplitude was determined using a 1 Hz sinusoidal waveform presented to the CE. The blocking threshold was determined by gradually increasing the LFAC amplitude during VNS induced bradycardia while simultaneously monitoring HR and BP. The threshold was defined as the current needed to cause a deviation that returned both towards their nominal values. At the same time, the potential across the blocking electrode was measured through the isolated voltage monitor to verify linearity of the voltage drop to the current presented (input/output). A linear input/output relationship was used as an indication that the maximum potential across the blocking electrode did not exceed the typical water window. For Pt-Black hook electrode, the typical reported water window range is −0.6 to 0.8 V versus Ag/AgCl while the typical reported range for PEDOT coated electrode is −0.9 to 0.6 V versus Ag/AgCl [26] expressed as half cell potentials.

To test the effect of LFACb, the vagal stimulus train and the LFACb waveform were presented in a regular continuous sequence as follows: 1—Pre Phase) ∼20 s baseline period of no stimulation, 2—LFACb Only Phase) ∼20 s LFACb delivered to the CE, 3—LFACb+VStim Phase) ∼20 s LFACb delivered to the CE and vagal stimulation to the RE, 4—Vstim Only) ∼10 s Vagal stimulation at RE or until BP falls below ∼50 mmHg. 5—Post Phase) No stimulation return to baseline. This test sequence was repeated on an average three times per animal followed by a control case. In the control case, the vagal stimulation and LFACb sites were swapped such that VStim was presented at the CE and LFACb was presented at the RE. Thus, the control cases would take into consideration the potential explanation for the apparent block to be an interaction between the LFACb waveform or electrode and the vagal stimulation pulse train or electrode [31]. The ECG, BP, the LFACb waveform, and voltage drop across the LFACb electrode were simultaneously recorded with each channel sampled at 10 kHz via a data acquisition system (USB-6212, National Instruments, Austin, TX, USA) controlled by an experimental data acquisition software (Mr. Kick III, Aalborg University, Denmark) to a PC workstation.

### 2.4. Data Analysis

The acquired data sets were analyzed using custom written routines in MATLAB (2016a, The Mathworks, Natick, MA, USA). The continuously acquired ECG and BP were segmented into five epochs corresponding to the conditioning sequence and identified as follows: Pre, LFACb Only, LFACb+VStim, VStim Only, Post. Based on the R to R interval of the QRS complex, an R-R rate and median R-R rate were calculated for each epoch. The normalized heart rate (Normalized HR) during each experimental epoch was calculated using Equation (Equation 1) as a percentage.
(1)$$Normalized\;HR\left(\%\right)=\left(1-\frac{\left[cond\right]-median\left(RRrate_{pre}\right)}{median\left(RRrate_{pre}\right)-median\left(RRrate_{vstimonly}\right)}\right)*100$$
where the difference between the median in R-R rate_pre_ and R-R rate_VStimOnly_ represents the maximum depression in HR, while the change in R-R rate, and [cond] represents the median R-R rate for each of the five epochs. This equation normalized the HR in the ‘Pre’ and ‘Stim Only’ epochs. ‘Pre’ was normalized to 100. Meanwhile, ‘Stim Only’ was normalized to 0. The normalized HR directly correlates 1:1 to the percent block in the ‘LFACb+VStim’ and ‘VStim Only’ epochs in both the test and control cases.

### 2.5. Statistical Analysis

The test case repetitions were combined for each electrode and two separate one-way ANOVAs were performed in R Studio (1.2.5042, R Foundation for Statistical Computing, Vienna, Austria) for each electrode type (cuff or hook) to determine if there was a significant difference between each of the five epochs of the test sequence. A two-way ANOVA was performed to determine the effect of electrode type and treatment (Pre, LFACb Only, LFACb + Vstim, Vstim Only, Post) on normalized HR. Lastly, the test and control case repetitions were each combined into a respective data set and a second two-way ANOVA was performed to determine the effect of experimental case (test or control) and treatment on normalized HR. Tukey post hoc tests were used to determine where, if any, the significant differences were located.

## 3. Results

The VStim trains induced an episodic reduction in heart rate which presented as an increase in the RR interval with dropped beats. As a result of the dropped beats, the BP also dropped. Figure 2 and Figure 3 are data presented from the same continuous experimental recording of a run. Figure 2 shows the filtered ECG vs. stimulation epoch, while Figure 3 converts the ECG to RR rate and plots the simultaneously measured BP. Focusing on the VStim Only epoch, one can see the episodic reduction in HR. The response time of the blood pressure due to vagal stimulation is relatively slower than the response time of HR. Additionally, BP can drift due to occlusion of femoral catheter, vasoconstriction, or vasodilation. Therefore, the heart rate provided a more reliable biomarker and measure of block. The change in RR rate is most clearly seen by examining the minima during vagal stimulation alone. Therefore, the RR rates were calculated and the local minima in rate associated with dropped heart beats were used to quantify the effect of the vagal stimulation without block.

Expanding out to the other epochs in Figure 3, the plot shows the change in ECG and blood pressure as a function of stimulation condition. Application of LFACb alone does not alter the ECG rhythm or mean blood pressure. This indicates that the LFACb waveform did not activate the vagus nerve. When LFACb is used in conjunction with vagal stimulation, there is very little to no change in either rhythm or blood pressure. This suggests that LFACb is blocking the effect of vagal stimulation, whose effect without LFACb can be seen in the fourth epoch, VStim Alone. Once LFACb is removed, the effect of the vagal stimulation is unmasked and there is a rapid disruption in the heart rhythm as well as the blood pressure. When vagal stimulation is removed, the heart rhythm returns to its initial state after a slight overshoot in heart rate and blood pressure. This is attributed to sympathetic rebound. The blood pressure trends follow that of the heart rate since it is coupled to the pumping function of the heart. Trial repetitions were spaced out so that the animal had returned to the same ‘Pre’ epoch before running the subsequent trial runs. Vagal stimulation alone or without block results in the complete stoppage of the heart if not discontinued.

The percent block was estimated using Equation (Equation 1), the normalized HR, but applying the value of the local minima of the RR rate as the marker. This is shown in Figure 4. The median of each epoch is then used to derive an estimate of the percent block and its variance for each epoch. The percent block represents the prevention of disruption to the heart rhythm. The absence of a RR rate depression during LFACb+VStim suggests that LFACb blocked the effects of vagal stimulation projecting to the heart. In this particular example, LFACb achieved a 100.8±3.3% block of the effects of vagal stimulation.

Interaction between electrodes, current sources or waveforms could be the source of the nerve block. As a control, the block and vagal stimulation electrodes were reversed. If interaction between electrodes or waveforms caused the observed block, swapping the application points of the two stimulation types should also result in a block in the ‘LFACb+VStim’ epoch. If not, then swapping will not effect a block in the ‘LFACb+VStim’ epoch and a depression of the heart rate should result. A typical result of the control case is shown in Figure 5. The calculated percent block in this control case was 2.9%. The effects of vagal stimulation were not blocked, strongly suggesting that block using LFACb is not due to an electrode or waveform interaction.

### 3.1. Hook Electrode

Table 1 summarizes the experimental parameters used and the percent block that was achieved when using a Pt-Ir bipolar hook electrode, electroplated with Pt-Black electrode to deliver the LFACb waveform. The average percent block among all experimental test cases was found to be 86.6±11.3%. In one case, the instrumentation had connection issues which prevented currents from increasing beyond 2.5μAp after being presented to the electrode. Despite the limitation, ∼60% block was achieved. In the control case, an average percent block of 7.3±26.3% was achieved during the ‘LFACb+VStim’ epoch. The negative percentage indicated that the R-R rate during ‘LFACb+VStim’ was below the normalized ‘VStim only’ epoch.

### 3.2. Cuff Electrode

Table 2 summarizes the experimental parameters used and the percent block that was achieved when using a bipolar cuff electrode to deliver the LFACb waveform. The average percent block was calculated to be 84.3±4.6%. Alternatively, the control case yielded an average percent block of 3.6±12.6% during the ‘LFACb+VStim’ epoch of the test sequence.

### 3.3. Statistical Analysis

One-way ANOVAs were performed to determine differences in the normalized heart rate versus conditioning epoch for the hook and cuff electrode data. The results indicated that there was a statistical difference within the five epochs of the test sequence (F(4,125) = 755.6, Pr (>F) = $2\;\times \;10^{-16}$ and F(4,70) = 2271, Pr (>F) = $2\;\times \;10^{-16}$, respectively). Tukey post hoc results showed that there was no significant difference between ‘Pre’ and ‘LFACb only’ in both the hook and the cuff (p-adjusted = 0.99, p-adjusted = 0.77, respectively) data sets. A subsequent Two-way ANOVA was performed to determine whether there was a difference between the results obtained using a hook or cuff electrode. This combined result represented in Figure 6 supported the hypothesis that there was no significant difference between cuff and hook ((F(1,195) = 0.046, Pr(>F) = 0.830)). Additionally, Tukey post hoc results indicated that there was no statistical significant difference between ‘Pre’ and ‘LFACb Only’ (p-adjusted = 0.98). This further demonstrates that there is no onset response associated with LFACb.

Two-way ANOVA statistical analysis results examining the effect of experimental case (test or control) and treatment shown in Figure 7. The two-way ANOVA results revealed a significant difference due to experimental case (F(1,310) = 229.6, Pr(>F) = $2\;\times \;10^{-16}$), a significant difference due to treatment (F(4,310) = 1607.5, Pr (>F) = $2\;\times \;10^{-16}$), and a significant difference between experimental case and treatment (F(4,310) = 229.8, Pr(>F) = $2\;\times \;10^{-16}$). Furthermore, Tukey post hoc results revealed that there were no significant differences in the ‘Pre’ vs. ‘LFACb Only’ epoch between and within the control and test cases. However, there was a significant difference in the ‘LFACb+VStim’ epoch between the control and test (p-adjusted = 0), as expected.

## 4. Discussion

The LFACb waveform at 1 Hz with current levels less than 200 µA_p_ was sufficient to achieve >80% block of the effects of the descending activity generated by pulsed vagal nerve stimulation (VNS). The experimental protocol systematically determined whether LFAC alone evoked any changes to the heart rate or blood pressure, and showed that there were no visible or statistical changes against a no-stimulation baseline when applied. It also tested whether LFAC blocked the onset of bradycardia when pulsed VNS was introduced after LFAC was applied. This test showed that on average 86% of the effect of pulsed VNS was prevented by LFAC. Although not systematically tested when LFAC was applied after pulsed VNS evoked bradycardia, we found that LFAC reversed bradycardia, which returned when LFAC was removed. This opens up the possibility that LFAC block could be applied in response to the onset of a functional biomarker change, rather than as a continuous preventative measure.

The maximum peak current that can be delivered without the hydrolysis of water is a definition of the ‘water window’. The LFACb waveform amplitudes are reported as current amplitude to peak of the sinusoid instead of amplitude peak-to-peak. Since the current is delivered from one contact of the bipolar electrode pair to the other, the to-peak currents and the associated to-peak potential across the two electrodes are the relevant maximum potentials. The negative phase of the sinusoid is equivalent to swapping the polarity of the electrode pair. Within the electrode pair, the relevant water window potential is the potential due to oxidation at the anode and the reduction potential at the cathode. Since potentials are measured across the two half cells with electrodes of the same material and geometry, the water window is approximately twice the absolute value of the potential to the smaller of the potential needed to hydrolyze water at the anode or the cathode. This point was estimated qualitatively by measuring the distortion from linearity of the voltage monitored across the electrode pair. Monitoring the linearity is directly related to the ability of the LFAC waveform to reverse the Faradaic reactions at low frequency impedance. The charge carrying capacity of our electrodes was enhanced greatly through the use of Pt Black and PEDOT based Amplicoat^®^ [32]. We monitored the linear input-output relationships between current and electrochemical cell potential-which is directly related to charge injection- as an indicator of the electrode reversible operation within the reported water widow ranges. The LFACb waveforms generated were kept well within the point where non-linearities occurred, keeping the potentials within the water window during the experiment. The blocking electrodes used did not cause any apparent injury to the nerve as the nerve was able to conduct and ‘VStim only’ elicited bradycardia following the removal of the blocking waveform. The blocking effects were immediate without onset activation as is associated with kHFAC. When the LFACb waveform was discontinued, the effects were instantaneously reversed. Each experiment could last several hours, and after the experiment, the electrodes were removed and there was no visible change in appearance.

The continuous bradycardia inducing VNS pulses were phased relative to the LFACb peak such that the LFAC peak coincided with a pulse of the train to maximize the effect of block. LFAC blocks only episodically when the current is above the threshold of block during the course of the sinusoidal waveform. Prior work in earthworm and canine vagus nerve [25,30] as well as in silico experiments [33] revealed that the block occurs only under the cathodic phase and the blocking electrode swaps from one contact to the other during a single cycle of the sinusoidal waveform. Thus, although the blocks are episodic, there are two blocking periods in one LFAC waveform cycle and there is no block at the waveform’s zero crossings. Given that in our test, continuous 25 Hz pulsed VNS evoked bradycardia is functionally blocked by the episodic LFAC waveform, block did not need to be continuous and disruption of the rate code was sufficient to achieve functional block. However, if complete continuous block is required, it can be achieved using multiple sets of electrode pairs in which the peaks of the LFACb waveform are phased in way that the duty cycle of LFACb approaches 100%, as shown in Horn et al. [25].

We are currently looking into the mechanism of LFAC block using an in silico model and examining the gating variables to identify its origin [33]. The model is suggesting that LFAC block where nerve conduction block is associated with a closed state inactivation of the Na_v_1.7 TTX sensitive voltage gated sodium channels [25] under the cathodic side of the bipolar electrode pair. LFACb blocks by inactivating the activation state variable m, and activating the inactivation state variable h. Whereas in DC block [4] there is direct activation of h without activating m. In addition, electrode interaction was rigorously ruled out by the use of a control. The control groups showed no block and provided evidence that the block effect observed was not due to an interaction between electrodes or waveforms.

Additionally, in silico models suggest a smaller caliber first block order, which is congruent with what is seen in DC block, suggesting that the fibers blocked were A-δ or unmyelinated C-fibers [25]. As shown by McAllen et al., the conduction velocities of cardiac vagal fibers that generate bradycardia upon electrical stimulation are between 3 and 15 m/s [34]. Needle recordings of neural activity during bradycardia revealed that the conduction velocity of the fiber generating bradycardia was approximately 0.6 m/s. While this evidence is preliminary, the conduction velocity calculated points to unmyelinated C or A-δ fibers [35]. These fibers are descending due to the vagal crush and are preganglionic in origin and lie rostral to the cardiac ganglion [36]. Therefore, they are likely A-β, A-δ or C fibers. At low frequencies enough current penetrates the nerve to result in a functional change in a biomarker, i.e., block bradycardia, while staying well within the water window. However, for larger nerve sizes, this may not hold. An important observation is that the effects of vagal stimulation aren’t completely blocked, indicating that some fibers were not blocked.

In comparison, kHFAC is generally in the range of 1–10 mA_pp_ (0.5–5 mA_p_) [7,22,37]. This results in higher power dissipation compared to LFACb. Larger power dissipation can result in a larger embodiment of a clinical stimulating device. Thus, LFACb has the potential to yield power savings compared to kHFAC block. DC block has been shown to work in the order of 50 µA to 7 mA [4], indicating that LFACb achieves block on the order of that required for DC block. However, unlike DC block, the current is reversed within each cycle and the delivered charge is recovered. The scheme used to deliver the LFACb waveform is similar to bipolar pulse stimulation. The LFACb waveform current is delivered from one contact of a bipolar electrode structure to the other contact. Like a biphasic charge balanced pulse, the current reverses itself during the course of one cycle. In one half cycle, one contact of the bipolar pair is the anode and the other is the cathode. In the other half cycle, the first contact becomes the cathode and the second the anode. As a measure of safety, the voltage across the LFAC bipolar electrode pair was continuously monitored to verify that the potential returns to zero after each run.

In summary, the application of low amplitude current sinusoidal waveform through a bipolar pair of electrodes generated continuous block of continuous 25 Hz pulse VNS evoked bradycardia. The block came without onset activation and was maintained as long as the LFAC waveform was applied. Although long term safety remains to be determined, the results suggest that LFACb could be a potential method for achieving block at DC block current levels but without the potential for reaction products, and without the onset activation seen with kHFACb, making it an attractive alternative to obtain functional nerve block for use in future subtractive neuromodulatory therapies.

## Figures and Tables

**Figure 1 sensors-21-04521-f001:**
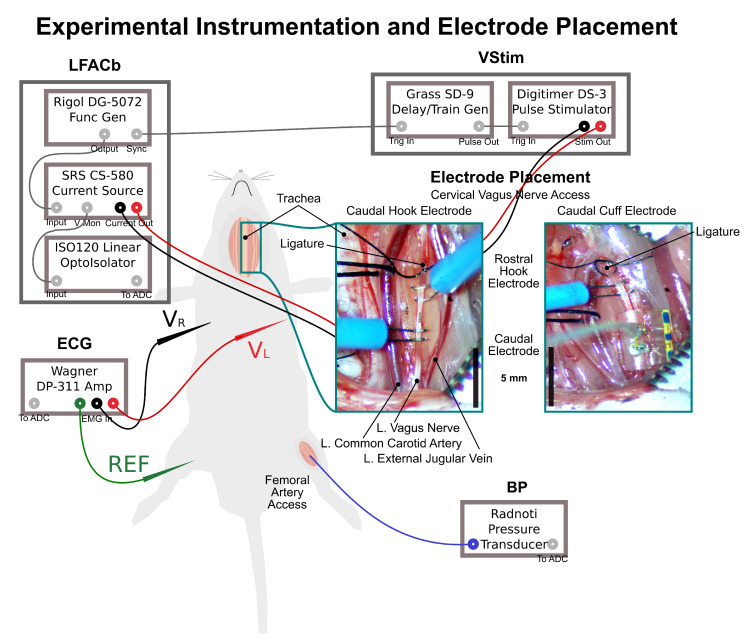
Instrumentation and electrode placed in the cervical vagus nerve access. The Electrode Placement panels show a magnified view of the cervical vagus nerve access in two cases with a 5 mm scale bar: (left) with the caudal electrode as a hook and (right) as a cuff electrode. The rostral electrode was always a hook electrode. The electrodes are shown wired only in the hook electrode panel for simplicity. A ligature was placed rostral to both electrodes to eliminate left cranial reflexes. The hook electrode panel shows the ligature tied, while the cuff electrode panel shows the ligature just prior to closure. Additionally, the left vagus nerve was crushed with forceps, as shown in the left panel, to further ensure that reflexes were excluded. The right vagus nerve was left intact and not crushed for stability.The signals recorded from the instrumentation were the amplified ECG, the voltage drop across the LFACb electrodes, and the arterial blood pressure.

**Figure 2 sensors-21-04521-f002:**
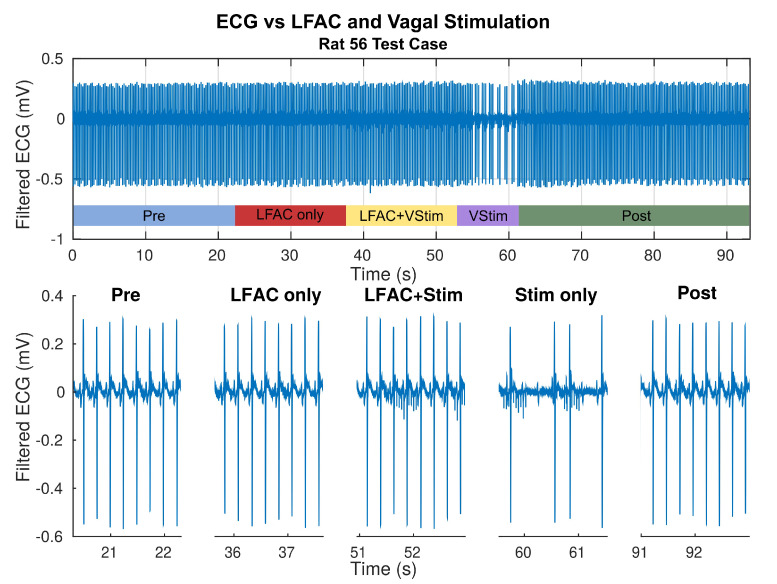
The effect on heart rate during a test sequence consisting of (1) Pre (no stim), (2) LFACb only, (3) LFACb and Vagal Stimulation delivered together, (4) Vagal Stimulation only, and (5) Post (Recovery, no stim or LFACb). The top panel shows a continuous recording of the bandpass filtered ECG during the five epochs. The bottom panels show 2 s samples of the ECG for each epoch. © 2019 IEEE. Reprinted, with permission, from [1].

**Figure 3 sensors-21-04521-f003:**
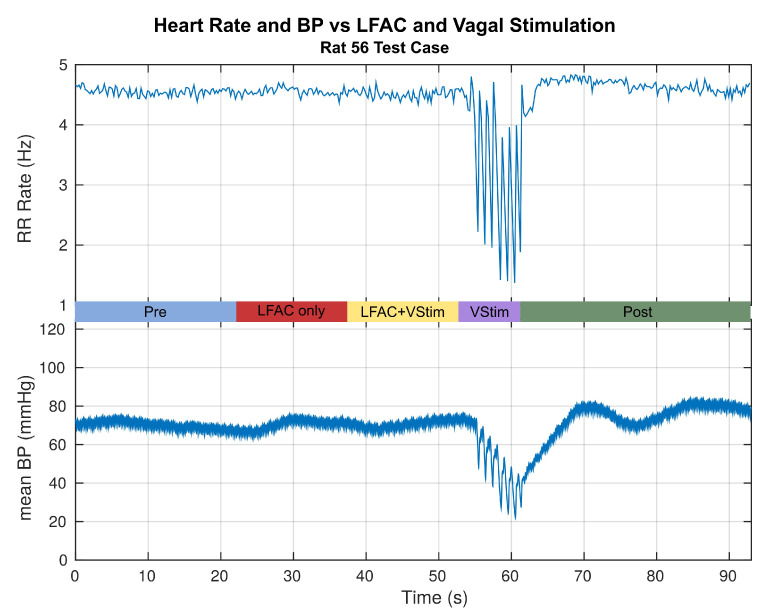
Effects of the typical test sequence on the RR rate of the QRS complex and mean arterial BP. The RR rate for this example is for the data presented in Figure 2. The HR and the BP show no change during LFACb and LFACb+Vagal Stimulation. This suggests that LFACb by itself does not activate fibers, and blocks the descending volley that elicits bradycardia and its accompaniment, hypotension. © 2019 IEEE. Reprinted, with permission, from [1].

**Figure 4 sensors-21-04521-f004:**
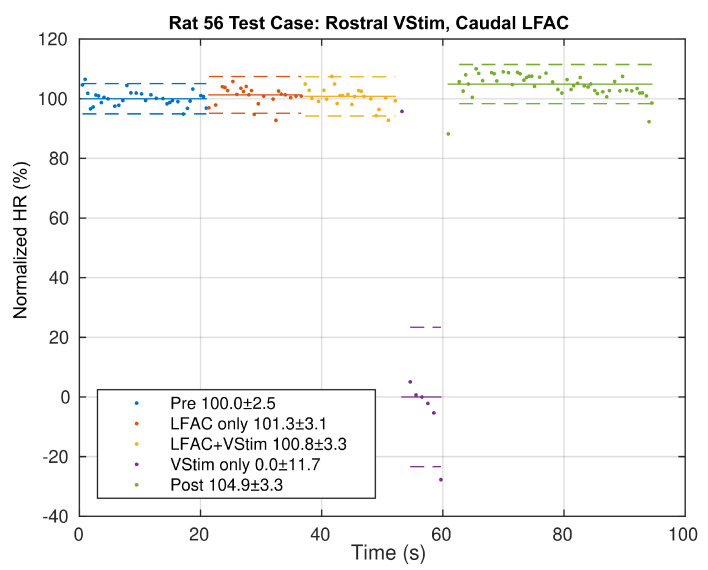
Example of Normalized HR (estimating RR rate derived percent block) as a function of epoch for the test case where vagal stimulation is presented rostral to the LFACb waveform along the nerve. In this example, 100.8 ± 3.3% block was achieved in the ‘LFACb+VStim’ epoch. A negative percent block, as in the ‘VStim only’ epoch, indicates that the data point was below the normalized median percent block in the ‘VStim Only’ epoch. The solid straight lines indicate medians and the dashed straight lines indicate one standard deviation. Legend values are % Block ± SD.

**Figure 5 sensors-21-04521-f005:**
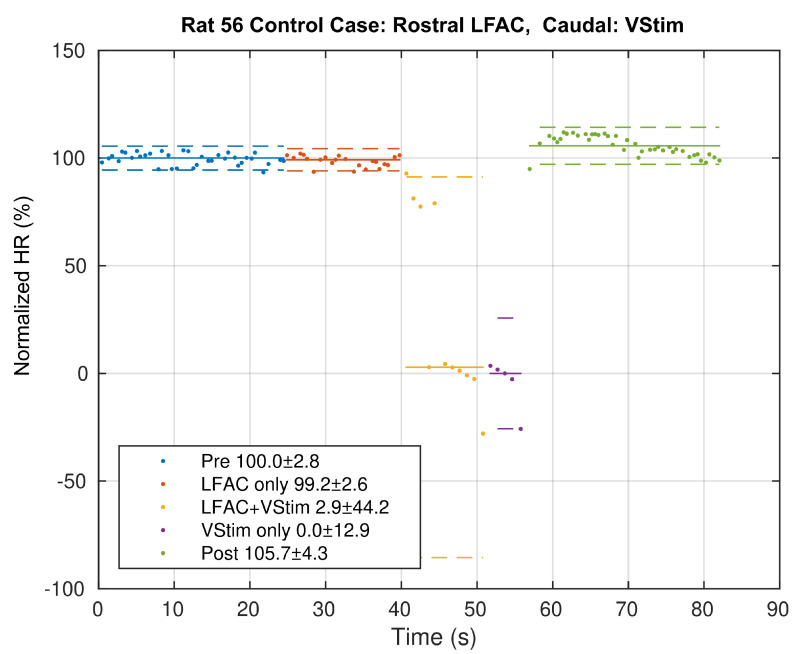
Example of the Normalized HR (estimating the RR rate derived percent block) as a function of epoch for the control case where LFACb is presented at the RE and vagal stimulation at the CE. Note that the ‘LFACb+VStim’ epoch does not show block, suggesting that the mechanism of action is not due to electrode or waveform interactions. The solid lines indicate medians and the dashed straight lines indicate one standard deviations. Legend values are % Block ± SD.

**Figure 6 sensors-21-04521-f006:**
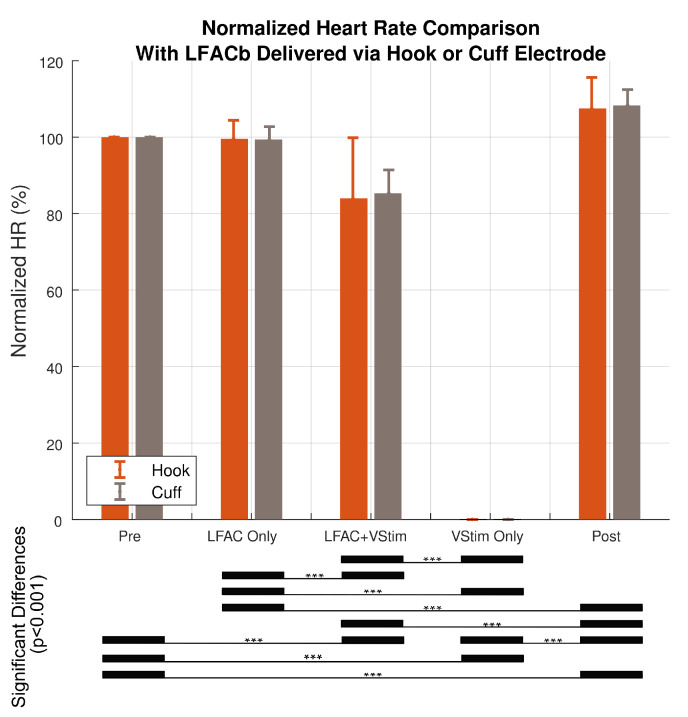
Summary of the effect of stimulation type and electrode type on normalized HR during for each of the five epochs of the experimental sequence (Hook N = 26, Cuff N = 15). There was no significant difference between the hook and cuff (Two-way ANOVA F(1,195) = 0.046, Pr(>F) = 0.830). The error bars indicate standard deviations. *** The significant differences ($p=2\;\times \;10^{-16}$) were only found between treatments.

**Figure 7 sensors-21-04521-f007:**
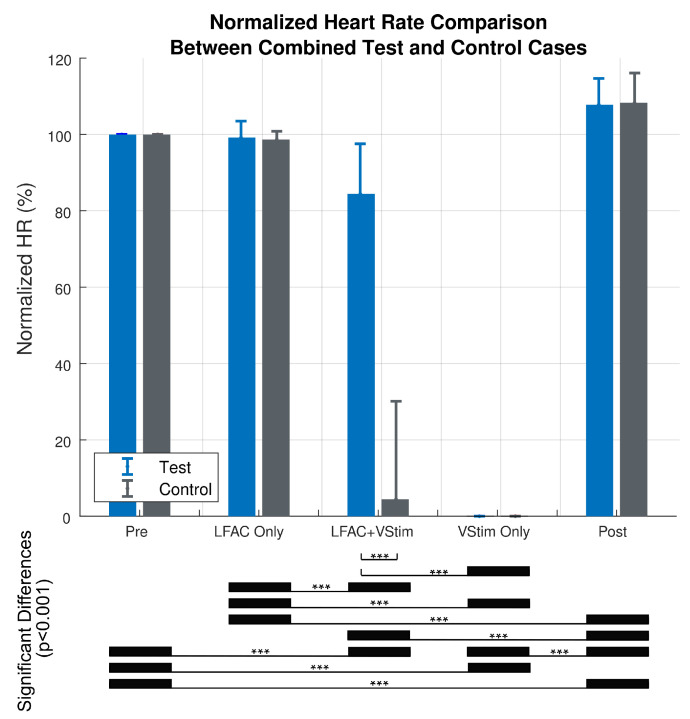
Summary of the normalized heart rate versus treatment for the Test (N = 41) and Control (N = 23) cases. The two-way ANOVA revealed a significant difference due to experimental case (F(1,310) = 229.6, Pr(>F) = $2\;\times \;10^{-16}$), a significant difference due to treatment (F(4,310) = 1607.5, Pr (>F) = $2\;\times \;10^{-16}$), and a significant difference between experimental case and treatment (F(4,310) = 229.8, Pr(>F) = $2\;\times \;10^{-16}$). There were no significant differences in the ‘Pre’ vs. ‘LFACb Only’ epoch between and within the control and test cases (Tukey post hoc). However, there was a significant difference in the ‘LFACb+VStim’ epoch between the control and test (p-adjusted = 0). The error bars indicate standard deviations. *** Indicates significant differences ($p=2\;\times \;10^{-16}$).

**Table 1 sensors-21-04521-t001:** Vagal stimulation and LFACb waveform parameters used in the set of 7 rats in this study. The conditioning waveform was applied via a hook electrode in this set. The average percent block amongst n = 7 experiments was found to be 86.2±11.1%. * Instrumentation connection issues did not allow currents >2.5μA. Nonetheless, ∼60% block was achieved.

Hook LFAC Block Experiments—Test Cases					
**Rat ID**	**Vagal Stimulation**			**LFAC Waveform**	**% Block**
	**PW (μs)**	**PA (μA)**	**Charge (Q)**	**Current (μA_p_)**	
Rat 46	100	270	0.03	160	83.0
Rat 50	1000	63	0.06	2.5 *	60.3 *
Rat 55	1000	29	0.03	100	83.1
Rat 56	1000	20.8	0.02	75	100.0
Rat 57	1000	19.5	0.02	75	68.1
Rat 58	1000	290	0.29	82.5	95.1
Rat 59	2000	180	0.36	50	87.7
**N**	**Mean**	**124.6**	**0.1**	**90.4**	**86.2**
**7**	**SD**	**120.0**	**0.1**	**37.7**	**11.1**

**Table 2 sensors-21-04521-t002:** Vagal stimulation and LFACb waveform parameters when LFACb was delivered via a bipolar cuff electrode. An average of 84.3 ± 4.6% block was achieved in the ‘LFACb+VStim’ epoch of the test sequence.

Cuff LFAC Block Experiments—Test Cases					
**Rat ID**	**Vagal Stimulation**			**LFAC Waveform**	**% Block**
	**PW (μs)**	**PA (μA)**	**Charge (Q)**	**Current (μA_p_)**	
Rat 66	2000	3500	7	55	83.0
Rat 79	100	78	0.01	65	78.8
Rat 84	1000	13.7	0.01	120	85.6
Rat 86	1000	22.4	0.02	195	89.8
Rat 91	1000	18	0.02	200	89.4
**N**	**Mean**	**903.5**	**1.4**	**108.8**	**84.3**
**5**	**SD**	**1731.2**	**3.1**	**64.2**	**4.6**

## Data Availability

The data presented in this study are available on request from the corresponding author.

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
