# Peer review of "A Reversible Low Frequency Alternating Current Nerve Conduction Block Applied to Mammalian Autonomic Nervesâ€"

_sensors, 2021, doi:10.3390/s21134521_

Round 1
Reviewer 1 Report
This paper demonstrates an interesting technique, but the text is very misleading. A complete rewrite of the introduction and discussion is needed to spell out the limitations with this technique and an electrochemical analysis of the electrodes needs to be added.
This introduction to this paper spends a lot of time discussing how DC can’t be used without special electrode chemistry and that multiple contacts would be needed to safely provide DC block and yet, that is exactly what this technique requires. The authors used specially coated electrodes to enhance the charge capacity of their electrodes. Although, there is no electrochemical data about the electrodes themselves other than the in vivo measurements. The waveform only blocks sporadically. In the discussion it is finally stated that in order to provide continuous block, multiple contacts would be needed. This is exactly how the charge balanced direct current (CBDC) carousel block referenced in the introduction works. And it is charge balanced just like the author’s waveform, in fact, it’s a part of its name!
It isn’t DC specifically that causes damage. The damage is caused by irreversible reactions, so a slow changing waveform potentially has the same problem. You need to specifically say that your waveform requires electrodes which allow you to expand the water window. You need to add something about the Shannon curve and charge/phase.
P3 line 94: Just because it is charge balanced, doesn’t mean that there is no accumulation of unbalanced charge. It may be that the levels you are using are OK with these electrodes, but both of these must be taken into consideration. This statement is oversimplifying the problem.
You need to include an electrochemical analysis of your electrodes. There should be a CV and an estimate of the charge capacity. Your technique of looking at linearity in vivo is fine, but the in vitro measurements are also needed to complete the picture.
In my experience, the effect of the vagal stimulation usually includes an overshoot and a settling period. Since the vagal stimulation is turned on after the block is applied, you can’t see where you are in this settling period. Also, DC block has a cumulative block effect which you may be seeing by leaving the block on for 10 seconds before applying the vagal stimulation. The protocol should have started with vagal stimulation, wait for it to settle, and then apply the block. Then you would see the effect of block over time and then may have seen the cumulative effect that occurs during DC block.
Your discussion about the mechanisms of DC block is oversimplified. DC block can be either achieved by depolarization/cathodic block(h gate blocked) or hyperpolarization/anodic block (m gate blocked). It’s just that since depolarization occurs at a lower value, when you attempt anodic block, the virtual cathodes exceed the block threshold before the true anode directly under the electrode. This doesn’t have anything to do with the interference issue that you tested for. The virtual cathodes would be closer to the electrode contacts than the distance between the stimulation and block electrodes-if that is what you were trying to say.
What this comes done to is that this is probably just DC block. On the vagus, you get away with it because the system responds slowly enough that you don’t see the gaps when your waveform goes to zero. Also, the DC block effect builds up over time as described in Vrabec, “A Carbon Slurry Separated Interface Nerve Electrode (CSINE) for Electrical Block of Nerve Conduction”, so you are probably seeing some carryover block in those periods as well.
Reviewer 2 Report
Authors present results in which low frequency blocking of nerve activity in vivo is demonstrated via the induction of bradycardia through stimulation of the vagus nerve. The results are very interesting and relevant to many fields of research, since illustrate an alternative to using blockade using DC or high frequency (kHz) electrical signals. The quality of the design and results is appropriate. I have two comments/suggestions: 1. How did authors prevented the possibility of a DC offset in their blocking signals? Please provide some explanation that tell readers that the effect is not caused by any DC current leaking into the circuit. 2. I am thinking that inverting the signals is not necessarily a fully convincing proof that there is not collisional blockade occurring. In my opinion, it may be possible to sync the signals in order to phase the collision properly in the the descending pathway (i.e. to the heart) and block the noxious stimulus. Was this already tested and ruled out?Author Response
Please see the attachment

Round 2
Reviewer 1 Report
See file
